

# TESSA: design and implementation of a platform for Situational Sea Awareness

M. Scalas[1], P. Marra[1], L. Tedesco[1], R. Quarta, E. Cantoro, A. Tumolo[1], D. Rollo[1], M. Spagnulo[1]

[1] Links Management and Technology S.p.A., via R. Scotellaro 55 - 73100 Lecce, Italy

*Correspondence to*: P. Marra (palmalisa.marra@linksmt.it)

## Abstract

This article describes the architecture of Sea Situational Awareness (SSA) platform, a major asset within "TESSA", an industrial research project funded by the Italian Ministry of Education and Research. The main aim of the platform is to collect, transform and provide forecast and observational data as information suitable for delivery across a variety of
channels, like web and mobile; specifically, the ability to produce and provide forecast information suitable for creating SSA-enabled applications has been a critical driving factor when designing and evolving the whole architecture. Thus, starting from functional and performance requirements, the platform architecture is described in terms of its main building blocks and flows among them: front-end components that support end-user applications and map and data analysis components that allow for serving maps and querying data.
Focus is directed to key aspects and decisions about the main issues faced, like interoperability, scalability, efficiency and adaptability, but it also considers insights about future works in this and similarly related subjects. Some analysis results are also provided in order to better characterize critical issues and related solutions.

## 1 Introduction

TESSA ("TEchnology for the Situational Sea Awareness") is an industrial research project under the National Operative
Program "Ricerca & Competitività 2007-2013" of the Italian Ministry for Education, University and Research. TESSA aims at strengthening and consolidating services belonging to operational oceanography in Italy and, in particular, in its southern seas. TESSA integrates both weather (e.g. wind, air temperature, precipitation, cloud cover, pressure), marine (e.g. wave height, period and direction) and ocean forecasts (e.g. currents, sea temperature) and analyses with advanced technological platforms that will allow an unprecedented dissemination of the environmental information for the "Situational Awareness"
at sea. Situational Awareness is a relatively recent concept and there has always been a lack of an agreed-upon definition because of the context-dependent nature of the concept (Sarter et al, 1991): Endsley et al. (2000) defined it as "the perception of the elements in the environment within a volume of time and space, the comprehension of their meaning and the projection of their status in the near future". Originally, Situation Awareness was an aviation term, but it has also been applied in many other safety critical domains (Stanton at al., 2001), like the maritime sector is.



Within TESSA, we meant Sea Situational Awareness as the capability to provide information about present and future sea conditions to support people during their activities at sea. This issue topic is strategically important for safer or optimal navigation, search and rescue, the assessment of the good environmental status of the marine ecosystem and the management of the sea territory. The lack of knowledge and awareness about sea conditions reduces readiness for reacting to emergencies

and protecting the marine environment, with large socio-economic impacts and damages. With the "Marine Knowledge 2020" communication (Green Paper, 2012), the European Commission outlined the importance of marine knowledge in helping EU Member States meet targets about employment, innovation, education, social inclusion and combat climate change as outlined in the "Europe 2020" strategy. Marine knowledge would provide the fundamental knowledge to facilitate the growth of a sustainable, job-creating "blue economy" in marine and maritime sectors by improving the competitiveness

and efficiency of industry, public authorities and researchers.

It has been estimated that marine data fragmentation and inaccessibility causes a loss of 300 million Euros per year, and that estimation does not take into account the future growth in marine economy and the subsequent increased demand for data. Opening up marine data allows new operators to enter the market while cross-systems interoperability allows businesses and academics to develop new products and services based on data from different sources and of different types. The impact

assessment could be of the order of 200 million Euros per year. In addition to this, uncertainty is one of the main problem of those responsible for designing offshore structures, for managing fish stocks or for designing protected marine areas. In example, it has been estimated that a 25% reduction in uncertainty in future sea level rise would allow public authorities responsible for coastal management to save approximately 100 million Euros per year.

The TESSA project was born from the collaboration between advanced oceanographic research, scientific computing and information technology groups. The project aimed at developing advanced technological platforms (web-based, multi-target and multi-channel) for the production of detailed information about sea conditions, at different complexity levels, and creating decision support systems for ship routing, extreme conditions early warning, environmental quality, oil spill movement forecasts and search and rescue operations.


This paper describes the main driving factors of the system and the most innovative solutions that have been implemented in order to satisfy the outlined requirements. Section 2 describes the motivations and objectives behind this system; section 3 presents the general architecture, including frontend and map-related infrastructure, while sections 4 describes the data analysis sub-system. Section 5 provides insights about the issues faced in implementing a scalable platform, with particular

emphasis on the map rendering issues. Finally, section 6 outlines conclusions and future works.



## 2 Motivations and objectives

The context of interest for sea conditions is wide, ranging from private activities like leisure (tourism, diving), sports and fishing, to institutional activities like environmental protection, search and rescue, intervention in case of oil spill and to business like safe navigation and offshore activities.

In that context, it becomes very important to properly and effectively deliver information about sea conditions: within TESSA project we translated this issue into the following objectives:

- to make information available everywhere and anytime: users would like to access information about sea conditions 24 hours for 7 days and also have data that are contextualized to the place where they are ("Tell me what are sea conditions in Otranto in the next three hours");

- to provide information about present and future sea conditions that are as easy as possible to understand. As an example, for services targeted to the large public, by endorsing easy-to-use user interface paradigms to convey scientific information and, at the same time, by adopting conventions (like naming of variables or of units of measurement) that are publicly recognized within scientific and professional communities;

- to provide information and service to both end users and service providers that will further transform sea conditions
data into other information or services;

- to transmit information and services via the more diffused devices (like smartphone or tablet);

- to provide easy to use services, in line with the modern web and mobile applications.

All those points have been considered as the main concerns in designing and implementing the SSA Platform, a
technological solution for delivering data and services for improved sea situational awareness. Those goals have been translated into the following system requirements:

- to transform data about present and future sea conditions into geo-referenced information like maps, graphs, and other additional knowledge not evident from raw data;

- to automatically download, process and publish environmental data every day;

- to provide different information formats, on the base of channel capabilities (i.e. different bandwidths between web and mobile) or device features (i.e. display dimensions), minimizing publishing and downloading time;

- to provide interoperable services, making them available through de-facto standards like Google Maps' Tile Map Service[1] (TMS) and standard OGC WMTS[2] protocols.

An example of an easy to use service is SeaConditions (Coppini, 2016), which provides ocean and weather forecasts for the Mediterranean Sea across different channels like modern web browsers and the most widely-available mobile platforms

---

[1] http://www.maptiler.org/google-maps-coordinates-tile-bounds-projection/
[2] http://www.opengeospatial.org/standards/wmts



(Android, Apple). By itself, SeaConditions, provides a custom User eXperience (UX) exposing its own business logic and allowing the user to access well-defined use cases. This is made possible by the availability of a general software stack, called the SSA platform, designed to be re-used in many situations as envisioned by the TESSA project (like displaying maps).

Hence, the SSA Platform is a software infrastructure that has been developed in the context of the TESSA project in order to collect, transform and provide forecast and observational data as information suitable for creating multi-channel applications like SeaConditions and VISIR (Mannarini, 2016). Data analysis and presentation with care for efficiency, reliability and interoperability have been main requirements driving the platform design and implementation. During the project lifespan,

several prototypes have been developed trying to overcome the challenges that have emerged during the project's timespan, mostly concerning the huge amount of data that have to be processed and provisioned. In this way, the architecture has emerged by a constant evolution driven by new functional and non-functional requirements, instead of being a complete up-front design.

## 3 General Architecture

Within the TESSA architecture, three main tiers can be identified: the client tier (e.g., mobile devices or web browsers) representing client applications (like SeaConditions), the Complex Data Analysis Module (CDAM) tier, hosting forecast and DSS-specific models and the SSA Platform, processing the forecast data from the CDAM tier and provisioning map data to the client tier. The SSA platform communicates with the CDAM by fetching weather and marine forecast data as needed and by managing job submissions on behalf of the DSS applications.

In a glance, the SSA Platform provides the building blocks for creating SSA applications, like standard services (e.g., the static and dynamic map services), hosting infrastructure (e.g., web server and messaging support), multi-channel availability (based on standard Internet protocols) and personalized user experience (e.g., sharing user preferences across several applications and channels).

In order to make the data flow serviceable, there are several specialized software components involved, each one performing a well-defined task within the processing chain. These components can be grouped in two macro-areas: (1) front-end components, hosting applications and user-centered data, and (2) map and data analysis components, dealing with serving maps and querying data.

Frontend components include a web portal, which hosts Decision Support Systems (DSS) like SeaConditions and VISIR; a message broker which enables communication between the DSS and their CDAM counterparts; and a user database hosting user preferences (like application settings and favourite places).




Map components include a download daemon which fetches data from the CDAM; a batch rendering system which performs initial ingestion within the system; a map service featuring tiled maps for forecast data; and a map server serving static maps like bathymetry. Data analysis is provided by software modules that allow for on-the-fly queries of forecast data, according to a variety of patterns. The greater part of these components has been developed ad hoc for TESSA, leveraging Open
5   Source software stacks and open Internet-based protocols wherever possible in order to maximize reuse of well-known and proven base packages.

**Figure 1. SSA Platform architecture (color key: blue for frontend components, green for map components, red for external modules or externally produced data)**



The Web Portal is also tasked with ensuring the right authorizations, so that users can access applications according to their given permissions. Frontend components are made available through an HTTP reverse proxy[3] which manages URL mapping from a single host name (e.g., www.sea-conditions.com) to the right service, like the forecast webservice, the GeoServer[4], the message broker and the web portal itself.

The Map service provides the HTTP endpoints for serving forecast and observational maps and related metadata (though a custom RESTful (Fielding, 2000) API[5], using standard protocols like WMTS and TMS. Each day, data within the SSA Data Storage are processed by a Map Pipeline which partially processes the most time-consuming map tiles: at runtime, the system will autonomously render tiles that have not been built earlier while still serving already available ones. In addition to serving tiled maps, the Map service also contains logic for querying raw data (like the series of values of the sea temperature

across a 4 days' timespan); an additional module, called ANSWER (detailed in a specific section), provides more complex data query features. An instance of GeoServer[6], a well-known software package for managing static maps in open formats, is also present in order to serve static map layers like bathymetry.

It must be underlined that, while the web user interface of each DSS application is being hosted within the portal, the

computation-intensive part is not. In fact, the DSS are, by design, split in two parts between: the user interface and the module running the algorithm, the latter being hosted within the CDAM tier. This deployment scheme is functional for reusing the computational resources of the CDAM cluster for running model software while freeing the SSA Platform for performing its own data analysis and map-related tasks. Splitting each DSS is dictated by the different requirements of each application, where actions like submitting a request and getting a response may involve several seconds or minutes. The

Message Broker makes easier to build application with such detached "submit-and-wait" logic: applications only have to assemble a message with their custom payload and queue them; the Message Broker will then track the request and ensure that the response from CDAM is then delivered to the client.

### 3.1 Frontend components

Frontend components deal with providing infrastructure services that allow hosted services to provide data to clients according custom web APIs. Decision Support Systems (DSS) are hosted within the portal while an HTTP Reverse-proxy hides the complexities of the underlying subsystems behind URLs based on a single hostname (like it happens for the www.sea-conditions.com).

---

[3] A reverse proxy is a web server fetching resources from other servers on behalf of some client: this is often used in IT departments in order to provide a unique facade (e.g., host name) which hides several other servers (as it happens within the TESSA architecture), thus improving systems changeability.

[4] http://geoserver.org/

[5] Application Program Interface, a collection of programmatic interfaces used for developing software applications.

[6] http://geoserver.org/





The Web portal is a web container, based on Liferay[7] technology and supporting the latest JSR Portlet JSR 286[8] standard, customized for the requirements of the TESSA project and hosting the applications, their web APIs and providing fundamental shared services like authentication, authorization and user profile and preferences management.

Frontend applications, like all DSS, are hosted within the portal as standard portlets, min-web applications that can benefit of the infrastructure provided by portal containers like security, data sources, and so on. Within TESSA, DSS applications, like VISIR, may have a heavy duty computational part that is hosted within the CDAM tier: a portlet's main purpose is to provide a user interface and collect inputs, package them according to specific formats and queue them to the computing engine. In order to this, a set of components has been designed and implemented to decouple the frontend and the computing

engine in order to reuse the latter across different client applications, like web and mobile. Other DSS, like My SeaConditions, instead, do not require any model computation and relay completely on services provided within the SSA Platform tier.

The message broker is a middleware component that provides a channel between clients (e.g., web or mobile DSS

applications) and the computation models, hosted within the CDAM tier. This allows client applications to host the user interface only, collecting input and presenting results while the actual computation is performed on a super-computing infrastructure that accesses both data and raw computing power.

In order for this model to work effectively, client applications assemble parameters into job requests and submit them to Message Broker; this in turn checks the validity of the request and the queues it to the CDAM receiver. A hook, in form of a

callback, is added so that the CDAM can notify job completion, either successful or not, including the data payload, without any need to perform continuous polling for results. Because the portal is already connected to a dedicated relational database server, data related to job requests are stored within a separate schema hosted within the same database server.

The Message Broker is not coupled in any way to any particular DSS and acts like a "store and forward" queue: it receives and stores requests, forwards them to the computing engine and waits for responses. At the end of the process, web and

mobile clients may retrieve and process such results, presenting them in their specific way (e.g., showing drawing symbols over a map or displaying a data table).

An additional feature is the ability to throttle requests: the message broker allows for setting a maximum number of pending requests for each user-service pair (e.g., each user can only make one request at a time for the VISIR DSS). This allows to enforce limits on unnecessary workload on the CDAM tier while also preventing stale data to accumulate: in any case, a data

cleaning policy can also be configured in order to purge stale requests (e.g., removing any successful or failed requests within 24 hours from their submission date).

---

[7] https://www.liferay.com/
[8] https://jcp.org/en/jsr/detail?id=286



## 3.2 Map rendering and provisioning

The map service is the software gateway to all the information stored within the SSA platform and it is designed to provide dynamic maps (like daily updated environmental data), static maps (like bathymetry) and data analysis functions (like simple and advanced data querying) to external systems and end-user applications. From a high point of view, it can be decomposed as a web API (RESTful web service) provisioning dynamic and static maps, a batch rendering system that constantly updates the forecast data, and a computing cluster that performs the actual rendering work.

The system delivers:

- tile-rendered forecast maps and associated metadata for the Mediterranean basin, in order to allow applications to create client-side mash-ups;
- data querying functionality, allowing applications to browse data across the available forecast time ranges and supported variables;
- on-the-fly rendering of map regions that have not been pre-rendered by the batch rendering system.

Data querying enables clients to ask for the actual values associated for given geographical coordinates and set of variables, with the ability to specify start and end limits and get arrays of values for further processing (e.g., displaying an XY charts).

A RESTful API is provided to client for querying for available maps and accessing the data browsing functions, according to the supported environmental data (see section 2.1).

In addition to dynamic maps, static maps like country boundaries and bathymetry, are also provided in order to ease the creation of meaningful mash-ups within clients.

Batch rendering system periodically fetches environmental data from data centers into a local SSA Platform Environmental Data Storage and triggers their initial ingestion within the system: the process also includes basic integrity checks and partial rendering of maps. The latter behavior means that the batch rendering system only pre-renders a limited set of map tiles, in order to save resources while still allowing for rapid responses for most frequently used maps. For section of maps that still have to be rendered, an on-the-fly rendering is triggered, queuing the task to the computing cluster. This allows for more efficient resource usage, requesting computations only when map tiles are effectively needed. Clients' requests are put on hold until the deferred rendering process has been completed. The resulting tiles are then stored within the map store so that following requests for the same tiles will be hitting the cache and get served faster.

Because of the great deal of computations required for rendering maps, the rendering system has been developed to scale horizontally by simply adding more machines. This is fulfilled by a computing cluster that provides the raw power for the rendering tasks, serving both the batch and on-the-fly scenarios.



Map tiles rendering is a natural candidate for parallel execution: the same tasks are to be iterated on different data sets over and over. Initial implementations used a thread-based parallelism: there was a single machine with several CPU cores and each core run a single tile rendering task at the time: up to 4 tiles may have been rendered concurrently before a final collect-and-store phase happened. Because many-cores machines become extremely expensive with the increase of the number of

CPU cores, scalability was achieved through adding one or more machines and manually configuring it to threat a particular dataset (e.g., one machine dedicated to sea variables while another to atmospheric ones). This was not easily to operate in production environment and introduced a single point of failure (an issue with that machine may bring down the entire system).

The solution consisted in implementing a distributed rendering model: a master node navigates the tile pyramid and generating all rendering tasks. The latter are then queued to a rendering grid composed of worker nodes that only have one single job: to render as many tiles as they can according to their CPU cores. Scalability is then achieved by adding as many worker nodes as required and then automatically made available to the system, without a restart being required; similarly, a node may be turned off without the rendering cluster falling apart, besides having less computational power available. Each

worker node accesses data from SSA Platform Data Storage, executes the rendering (employing different techniques as required) and returns the result (consisting of image bytes plus metadata). Tasks are defined by descriptors assembled by clients (e.g., the dynamic maps service or the batch rendering system) and queued to the cluster. A built-in load balancing feature ensures that tasks are dispatched in a roughly fair manner across different nodes, so that no single node is particularly overloaded. We integrated the Hazelcast[9] grid computing middleware, which provides infrastructure for coordinating tasks

among multiple processes and machines and built on top of it our rendering pipeline.

Operational running of the system is allowed by the means of system logs and mailing: when issues happen (e.g., unavailability of data, invalid or corrupted files, internal errors, and others), personnel can investigate the problem and operate in order to fix it (like, re-running rendering jobs for variables that were not available at an expected time).

Supporting operational teams has been crucial in order to improve the final availability of the platform services to the final users and we further think of improving it by collecting statistics in order to formulate realistic service level agreements (SLA) with external users.

## 4 Data analysis

In addition to map rendering, the SSA platform also provides services for querying the data, currently providing a sequence

of environmental values for a given variable. As improvement to data analysis, a sub-system (called ANSWER) for

---

[9] http://hazelcast.org/



searching time slices matching specific conditions has been implemented (e.g., winds and sea currents matching some externally provided thresholds) and made available to external systems (like MySeaConditions (Coppini, 2016)).

## 4.1 Data queries and statistics

The simplest data query involves fetching values for a given variable from each NetCDF file, a process that may require a
few seconds before completing, depending on the current system load and complexity of the involved computations, which depend on the particular variable. In order to speed the entire computation up, a data files indexing is performed and consulted for checking each file: read is performed using Unidata's NetCDF-Java library[10] once the correct file has been identified.

The concept of a tile as a pre-defined geo-referenced bounding box has been used for data analysis too: these "data tiles" can
be used for relatively fast execution of statistics computations (like average, standard deviation, minimum and maximum values). Web APIs, either custom TESSA-defined or OGC's standards, make these features available to external clients, like SeaConditions. A significant limitation of the current implementation is that data queries are performed sequentially, taking no advantage of the rendering grid: this is something that we plan to improve in the future since the grid-based parallelism has been found to be a major asset within the platform and should be exploited in other scenarios.

In comparison, ANSWER, instead, provides more advanced features and it has been implemented in matching conditions for a given geo-coordinates point while a prototype has been developed for analyzing entire areas in search for matching conditions (e.g., most favorable places for sailing in the next few days): the latter is presented in the following sub-section.

## 4.2 ANSWER

ANSWER (Algorithm lookiNg for arbitrary conditions of Sea and WeatheR) is a module for verifying that arbitrary weather and/or marine and/or oceanographic conditions occurs in delimited areas of interest and time intervals. It's tightly integrated within the Map components, complementing the map infrastructure with advanced data query features.

ANSWER implements an algorithm which allows users (like MySeaConditions) to model profiles of customized thresholds by specifying geographical, desired weather and/or marine conditions and a forecast range (e.g., next 72 hours). Based on
these settings, the forecast data are analyzed in search for valid matches. This processing is based on forecast data for the Mediterranean Sea, that is the NetCDF files produced as part of the TESSA project.

There are two main scenarios that are enabled by ANSWER:

- fixed region: the user selects a Region of Interest (ROI) and defines a set of thresholds for one or more forecast

---

[10] http://www.unidata.ucar.edu/software/thredds/current/netcdf-java/



variables (i.e. "wind speed over 10 knots and air temperature over 20 degrees in Otranto"). The algorithm looks for time intervals in which user conditions will occur within the ROI;

- fixed time: the user selects a time range and defines a set of thresholds for one or more forecast variables (i.e. "wind speed over 10 knots and air temperature over 20 degrees in next 3 days"). The algorithm looks for places where user

conditions will be verified.

In order to do this, ANSWER computes selection and intersection operations (algebra) across multiple layers of variables. In a typical scenario the user chooses the variable(s) and related thresholds (greater than, lesser that, within), then ANSWER fetches from the SSA Data Storage the NetCDF files, extracting the involved portions and checking the values one geo coordinate (latitude and longitude) at a time.

The algorithm then assigns a value "1" if the condition is matched for the given value, or "0" if external to the wanted range: a new NetCDF file is then created, containing a matrix of 1's and 0's for each given point within the bounding box. This is the behavior of the selection operator. When only one variable is involved, the resulting NetCDF file is the final output of the Algorithm (e.g., it may get displayed to the user). If more than one variable is involved, instead, the process of selection is to be repeated a number of times. In that case a matrix of 1's and 0's will be produced, one for each variable.

The final result is then computed as an iterative intersection of all the matrices: the final result will be a new binary file, with "1" where all user defined conditions occur (a logic "AND" operation).

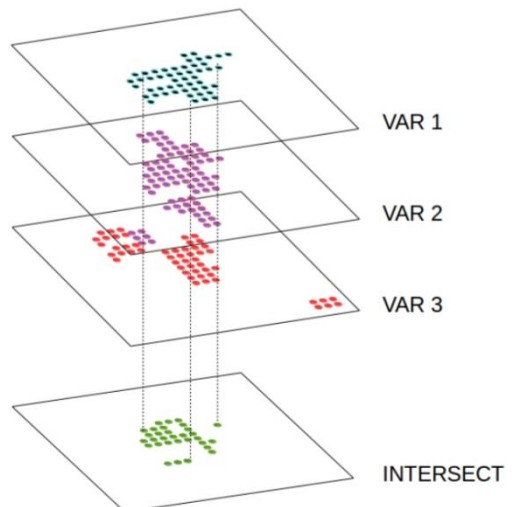

**Figure 2. Example of intersection between three variables**



Two prototypes of ANSWER have been implemented, in GrADS and NCL, in order to evaluate performances: GrADS has proved to be faster while providing. For example, considering an area roughly equivalent to a zoom level 5 tiles, computing conditions for five different variables requires, on average, about 0.3 seconds in GrADS in comparison to 1.3 seconds required by NCL.

### 4.3 Sample scenario

Let's suppose that the user is a surfer and is interested in the following thresholds for wave height and wind speed and direction:

1.  wave height: comprised 0 m and 0.5 m
2.  wind intensity: comprised 1.5 m/s and 3 m/s
3.  wind direction: ranging from NE to SE

In addition, let's also suppose that the user is interested only in Apulian coasts (southern Italy). ANSWER will first act by means of its selection operator that will produce NetCDF files and images, by highlighting places where user conditions are verified for each variable.

At this stage, the intersection operator will overlap results in order to identify ROIs that correspond to user defined conditions. Figure 4 helps visualizing the process by showing areas that are found to be matching wave height, wind direction and wind speed (respectively images a, b and c) during the selection phase. Intersection is then performed iteratively between matrix "a" and "b" (resulting in matrix "d") and, finally, between matrix "d" and "c". Figure 4e represents the final result.



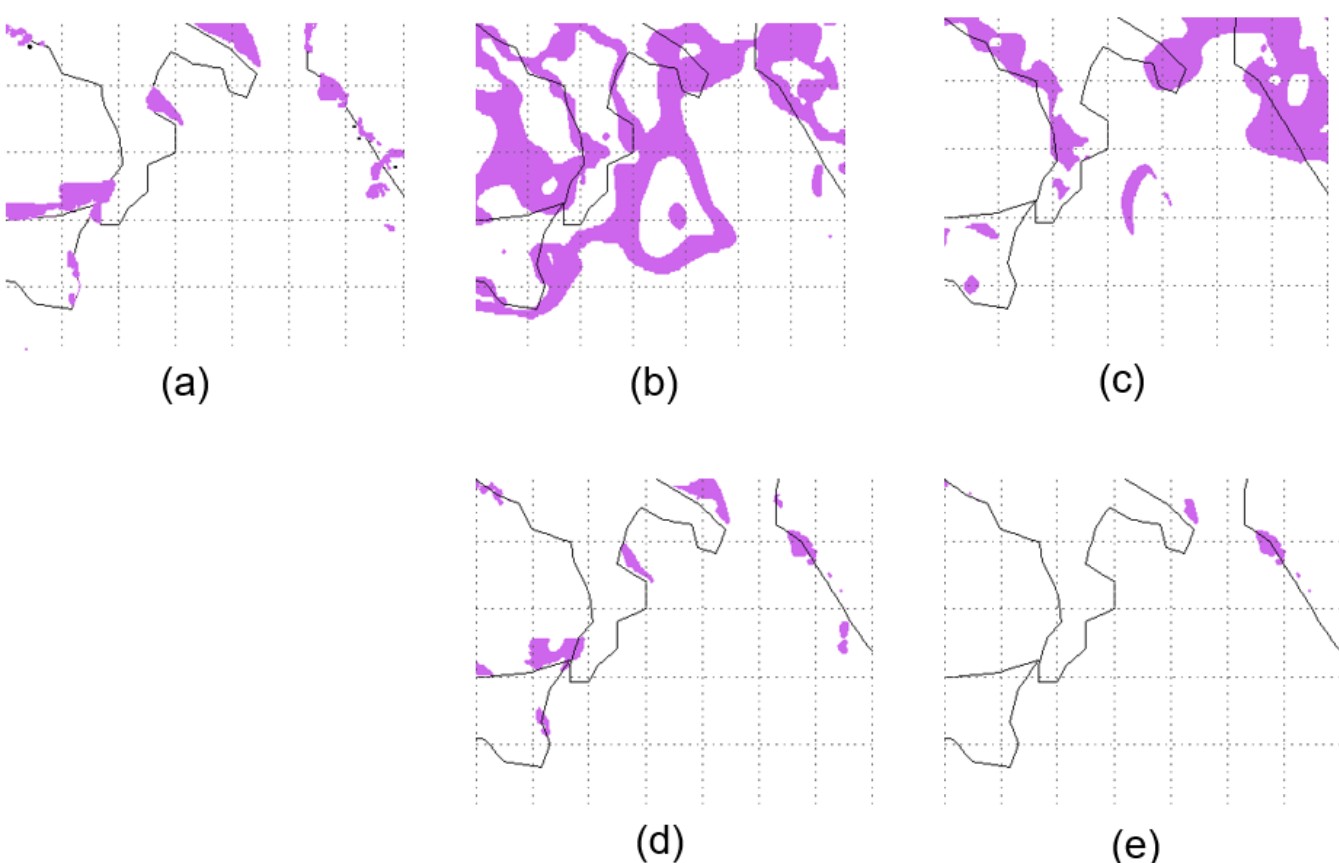

**Figure 3. Selection operator: sample areas matching the conditions during selection (a,b,c) and intersection (d,e) operations**

## 5 Discussion about map rendering and provisioning

5  Continuously producing and publishing maps presenting environmental data is a core business process of the SSA platform and the tile-based maps have already been used in other cases, like (Tarquini, 2008). A lot of effort has been put in the implementation of a system that can manage large amount of scientific data in reasonable time for daily updates while still being as efficient as possible about the usage of computational resources.

### 5.1 The map creation process

10  Maps are collections of small image tiles: in TESSA, maps are defined for each pair *<Variable, Date>* where *Variable* may be any environmental variable, like Sea Currents or Winds, and *Date* is any 3- or 6-hour time slice within the 5-days forecasts considered by the system. Tile-based rendering of maps is a common technique, used by software like Google



Maps[11] , that allows distribution of very large maps at different scales as a set of 256 pixels-wide images, greatly improving efficiency in bandwidth and client resources usage. In facts, the current working set a user is displaying within a single screen is very small and it makes sense for clients to only load and present this limited set. Several, predefined layers have been defined, called "zoom levels", and each with its own meters-to-screen resolution ratio: for example, zoom level 10 has

8 Kilometers per 90 pixels, zoom level 11 has 4 Kilometers for 90 pixels and so on. At all effects, each increased zoom level effectively doubles the resolutions and quadruples the amount of tiles that have to be rendered and provisioned: the set of tiles for a given map is called a "tile pyramid". In TESSA, each tile is either a color-shaded tile or a vector (e.g., arrow) tile, both being bitmaps representing different aspects of a single environmental field (like wind intensity and direction).

Usage of raster images simplifies clients that only need to fetch and present simple images: this is a trade-off which moves much of the rendering effort of the server-side (thus, requiring it to be adequately sized). Additionally, issues with the quality of the raster-based pictures have been found, like excessive pixel artifacts, lack of smoothness or excessive image blur (which is particularly evident in some cases, like stream lines). During the years, several formats[12] have been defined but none has been affirmed, for creating vector tiles, textual-representation of the geometric objects that have to be displayed

instead of the actual pixels to be displayed on the screen. In TESSA, we explored vector-based tiling but found no easy way to implement and no available client that may have been able to display it: rasterized tiles have then become an enforced choice while vector-based tiles may be explored again when the technology (both server- and client-side) will be more mature. Thus, raster-based maps, with all their limitations, have been found to be the best way for serving maps across the widest range of clients: yet vector-based tiles will probably be the future since it is gradually getting more wide-spread

usage, even in professional GIS software[13].

Conventions for indexing the images belonging to tile pyramid has been defined in the past by different vendors, like Google and Microsoft, and it has then been standardized as Open Geospatial Consortium (OGC) as OpenGIS Web Map Tile Service (WMTS) 1.0.0 specifications[14]. The dynamic maps service historically provides an implementation compatible with Google

Maps and Apple Maps, de facto standards for web and mobile platforms, but also provides an experimental WMTS service for improved interoperability with external GIS software. The map tiles repository uses its own internal storage and each different public API for accessing maps translates external indexing standards to the one used internally (which essentially maps on the Operating System's file system).

---

[11] https://www.google.it/maps
[12] http://wiki.openstreetmap.org/wiki/Vector_tiles
[13] In example, ESRI announced in February 2016 support for vector-based tiles in their ArcGIS system. See https://blogs.esri.com/esri/arcgis/2016/02/18/arcgis-10-4-is-here/ for more information.
[14] http://www.opengeospatial.org/standards/wmts



Within TESSA, the process of building a single tile is quite complex but can be abstracted as a fetch, process and store cycle that has to be repeated for each tile belonging to a single map (and, in turn, for each map that has to be rendered each day). The actual rendering is delegated to running scientific tools in batch mode (e.g., providing a script in order to get an image plot), like Grid Analysis and Display System[15] (GrADS) or NCAR Command Language[16] (NCL), or to custom rendering

routines (e.g., for arrows representing waves and winds) that is run in place of an external process. Usage of existing and well known tools greatly eased the implementation of the system, leveraging competencies across the working teams while also maintaining the scientific accuracy of the output; custom rendering has been introduced in order to solve rendering issues (e.g., pixelating artifacts) with  GrADS and NCL that could not be worked around.

## 5.2 Rendering algorithm

The algorithm that has been devised is a parallel, recursive tree traversal of the whole tile pyramid (see Figure 2a), starting from zoom level 5 up to a maximum zoom level[17] (other levels are not ignored): for each visited tile, the process of fetching data, rendering and storing the resulting images is performed. Initial seeding set is composed of 6 tiles that encompasses the entirety of the Mediterranean Sea at zoom level 5 and the amount of tiles to be rendered quadruples at each new zoom level: each single map can then require several thousands of maps tiles in order to be completely rendered:

$$MapTiles = \ 6 * (4^{i-5}), i = 5..11$$

The typical amount of map tiles consists of more than 32-thousand tiles (for each single map), multiplied by 29 time-slices (e.g., 29th Sept 2015 at 15:00, at 18:00 and so on) and multiplied by nine (the number of different forecast variables): this gives a grand total of about 8 million and half tiles that have to be generated each single day. In order to achieve the goal of

efficient and timely publishing, several optimizations and design solutions have been devised.

---

[15] http://cola.gmu.edu/grads/
[16] https://www.ncl.ucar.edu/
[17] The maximum zoom level it depends on the data resolution: the higher the resolution, the higher the maximum zoom level. In example, sea temperature (150 meters resolution) can be rendered up to zoom level 11 while precipitations (25 km resolution) can be rendered up to zoom level 9 (image quality starts to degrade after this level).



**Figure 4. Tiled map generation**

The algorithm starts generating rendering task for each single tile and its four child tiles until the maximum zoom level is

reached or the tile is filtered out (e.g., it is entirely on land while the variable being plot is a marine one). Filtering by land-sea mask enables to avoid about thirteen percent of the entire amount: the time necessary to compute if a tile is completely land-based (or marine-based) is negligible in comparison to the amount of plotting an empty tile: more than two orders of magnitude. The speed up has been made possible because the lookup table is completely available in memory during the rendering process.

Each tile is rendered according to a predefined set of steps:

1.  GrADS or NCL scripts are instantiated according to the specific variable at hand and limiting the data area to the coordinates related to the specific tile;

2.  the script instance is executing according to the specified plot engine, producing an image representing the environmental field in the given area;

3.  image is post-processed in order to make it of the correct size and proportions.



Introducing a new variable within the system required that custom rendering scripts had to be written and then parameterized once the wanted result is achieved: these script templates are stored and then retrieved at rendering time, with their placeholders being replaced with the actual values (e.g., NetCDF[18] file names, geographic bounding box,). The resulting

script is then given as input to NCL or GrADS and, if executing is successful, the resulting image in post-processed. Post-processing may involve re-projection of the image according to Mercator coordinates (used by tools like Google Maps), scaling to 256 pixels-wide images (rendering usually uses a bigger resolution and then scales it down in order to smooth the image) or further reducing the file size by additional data compression.

The entire map rendering process can be quite time consuming and it has been found to be directly related to data resolution (e.g., 150 meter files take longer to process that 25 Km data files), the area to plot (larger areas involve a larger number of data to be analyzed and thus greater processing times), number of files involved and computations to be performed (e.g., computing vector direction and magnitude takes considerably longer than just plotting the data read from a given file).

At the beginning, requirements dictated only up to zoom level 8 and the whole rendering process happened during the night in a "batch mode" on single node: all data files were processed and the related map published in about four hours. With the increase in the number of zoom levels and data resolutions (e.g., waves at 150 meters), rendering every single tile was no more feasible and a different solution had to be found: on-demand rendering.

While tiles at zoom level 5 may require up to 3.5 seconds to render on average (worst case scenario for the sea currents
variable), tiles at zoom level 10 only require 70 milliseconds on average: the latter timings still apply for additional zoom levels, meaning that we hit the time required to start the external tool (GrADS or NCL) and store the resulting image, which can't be further reduced.

Analysis was performed about each image transfer over the Internet and found that each image was transferred from the
platform (production environment) to the client (a web browser) in about 225 milliseconds (with about 30% variance depending on server load). Additionally, has been envisioned that rendering every single tile of every time is a waste of resources since not all tiles are going to be browsed by users This has paved the way to the idea of "on the fly rendering" from a given zoom level onwards instead of upfront rendering: from zoom level 7 or 8 (depending on each variable because of different resolutions), on the fly rendering is default setting without any noticeable effect on the user experience. In order
to minimize the rendering time for topmost zoom levels, what was initially the nightly batch rendering of maps is now the initial "rendering seeding" phase before the on-demand rendering steps in.

---

[18] Network Common Data Format, http://www.unidata.ucar.edu/software/netcdf/





## 6 Conclusions and future works

This paper presented the main goals and design solutions that have been faced in the evolution of the SSA Platform. The main goal behind it is to provide a common infrastructure for supporting the creation of application mash-ups that increase the sea situational awareness of a vast plethora of users. The Platform has evolved considerably during the project, being

refactored several times in order to match the increased data sizes and performance requirements needed to match daily availability to end-users. Scalability, availability, interoperability and efficiency in resource usage have been primary concerns during the design.

In must be considered that current implementation still has room for additional improvements and refinements like grid-

based data queries, client-based rendering using vector tiles instead of raster images and better health monitoring during the operational running of the system (which is critical in production environments and high-availability environments). These aspects will be part of future works, in addition to further improving interoperability with professional visualization systems and providing forecasts as web APIs that can be used by third parties for creating completely new applications.

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
