# Peer review of "TESSA: design and implementation of a platform for Situational Sea Awareness"

_Natural Hazards and Earth System Sciences, 2016_

## Referee Comment (RC1) · Anonymous Referee #1 · 25 Jul 2016

Review of **"TESSA: design and implementation of a platform for Situational Sea Awareness"** by Scalas et al. MS No.: nhess-2016-166

The paper describes the design and implementation of a platform for Situational Sea Awareness. Authors provide a detailed description of both the architecture and related services, in a well-structured paper. However, only an applicative scenario related to the ANSWER service, which is just a small part of the whole system, is reported. My recommendation is to revise the manuscript before it is considered for publication. The following changes are suggested.

(1) The authors did not report an analysis of the state of the art. I suggest to add a reference to other systems (if they exist) which provide analogous services or a subset of them, also discussing which are the main advantages of the TESSA platform. Otherwise the novelty of the TESSA platform should be emphasize, if any other similar systems can be found.

(2) I have two comments on the SSA architecture, shown in Figure 1:

- it is not very clear the data flow among client layer, DSS applications, Message Broker and DSS Computation Engine. Some DSS Applications (e.g. VISIR) has both the application and the computational components. If I have well understood, these two components should be interfaced through the Message Broker. However the figure does not show an explicit link between DSS Application and Message Broker, as well as between the Message Broker and the DSS Computational Engine
- it is unclear where the hardware resources are located in the system architecture.

(3) The term SeaCondition is used to refer both to the application client and the DSS application. I suggest to use different names for the two components

(4) In section 3.2, the hardware system used for the map rendering should be described, discussing also the time-to-solution for both the pre-rendering and on-the-fly rendering. Even if authors guarantee on the system scalability, experimental results are not reported.

(5) The affiliation of two authors is missing

(6) It is not clear if the acronym SSA stays for "Sea Situational Awareness" or "Situational Sea Awareness". Somewhere in the paper it seems to stay for "Sea Situational Awareness", in other points for "Situational Sea Awareness".

(7) At page 4 (line 15) the authors use the ambiguous expression "mobile devices or web browsers". Indeed, mobile devices seem to refer to the hardware clients while web browsers to the software clients.

(8) There are some mistakes in the text:

- page 8 line 16 'charts' instead of 'chart'
- page 8 line 18 reference to section 2.1 which is missing
- page 9 line 26 'Service Level Agreements' instead of 'service level agreements'
- page 10 line 20 there is a mistake in the ANSWER acronym: it is not clear what the 'E' stands for
- page 10 line 21 'occurs' instead of 'occur'
- page 10 line 30 'of' instead of 'Of'
- page 11 caption figure 2 'between' instead of 'among'
- in the text, references to figures 2, 3 and 4 are wrong or missing

---

## Referee Comment (RC2) · Anonymous Referee #2 · 28 Jul 2016

The manuscript is a report from a funded project with a broad spectrum of activities. It includes issues ranging from safe navigation to environment protection, tied by the sea element. I would appreciate to find a clear statement of the research goal or research questions. Above all I miss a related work section. How can I know whether the contribution is novel without a comparison with related work, providing evidence that the project is beyond the state of the art? Another important drawback is the lack of feedback from real users. Is there anyone who has used the platform? As a suggestion, manuscript might focus on the actual innovation aspects. A paper written as a full technical description tends to be boring. Another suggestion is to go through a careful proofreading, possibly by an English native speaker. Minor comments: DSS should be expanded the first time it is used as an acronym not later. Figure 1 should be explicitly mentioned in the text.

---

## Referee Comment (RC3) · Anonymous Referee #3 · 16 Aug 2016

The paper has one major issue to someone like me who's not knowledgeable in the filed: with no related work provided, how can one judge the scientific advance, if any?

The application may be unique, but there has been a considerable amount of previous work on: - information broker that provide unified access to a federation of different information source, typically leveraging ontologies - GIS systems in general

As per the definition of situational awareness, authors should also refer to the following paper M. R. Endsley, "Toward a theory of situation awareness in dynamic systems: Situation awareness," Human factors, vol. 37, no. 1, pp. 32–64, 1995.

Minor: - Sect 2: 24 hours per 7 days –> 24/7 (short) or 24 hours a day, 7 days a week (long)

---

## Author Comment (AC1) · 30 Sep 2016

**TESSA:** design and implementation of a platform for Situational Sea Awareness**

M. Scalas1, P. Marra1, L. Tedesco1, R.Quarta2, E. Cantoro2, A. Tumolo1, D. Rollo1, M. Spagnulo1

1Links Management and Technology S.p.A., via R. Scotellaro 55 - 73100 Lecce,Italy 2Fondazione CMCC - Via Augusto Imperatore 16, 73100 Lecce, Italy

Correspondence to: P. Marra (palmalisa.marra@linksmt.it)

**Abstract**

5

This article describes the architecture of Sea\_Situational Sea\_Awareness (SSA) platform, a major asset within "TESSA", an industrial research project funded by the Italian Ministry of Education and Research. The main aim of the platform is to 10 collect, transform and provide forecast and observational data as information suitable for delivery across a variety of channels, like web and mobile; specifically, the\_ability to produce and provide forecast information suitable for creating SSA-enabled applications has been a critical driving factor when designing and evolving the whole architecture. Thus, starting from functional and performance requirements, the platform architecture is described in terms of its main building blocks and flows among them: front-end components that support end-user applications and map and data analysis

15 components that allow for serving maps and querying data.

Focus is directed to\_key aspects and decisions about the main issues faced, like interoperability, scalability, efficiency and adaptability, but it also considers insights about future works in this and similarly related subjects. Some analysis results are also provided in order to better characterize critical issues and related solutions.

**1** Introduction**

- TESSA ("TEchnology for the Situational Sea Awareness") is an industrial research project under the National Operative 20 Program"Ricerca & Competitività 2007-2013" of the Italian Ministry for Education, University and Research. TESSA aims at strengthening and consolidating services belonging to\_operational oceanography in Italy and, in particular, in its southern seas. TESSA integrates both weather (e.g. wind, air temperature, precipitation, cloud cover, pressure), marine (e.g. wave height, period and direction) and ocean forecasts (e.g. currents, sea temperature) and analyses with advanced technological
- platforms that will allow an unprecedented dissemination of the environmental information for the "Situational Awareness" 25 at sea. Situational awareness is a relatively recent concept and there has always been a lack of an agreed-upon definition because of the context-dependent nature of the concept (Sarter et al, 1991): Endsley et al. (2000) defined it as "the perception of the elements in the environment within a volume of time and space, the comprehension of their meaning and the

Commento [MS1]: Review #1 -Cantoro and Quarta doesn't have any affiliation at present: CMCC was the affiliation they had when they were involved in TESSA project.

Commento [MS2]: Review #1 - Point 6: With an exception for the TESSA acronym, the expression "Situational Sea Awareness" has been thoroughly corrected in "Sea Situational Awareness" within the article.

Commento [MS3]: Review #1 - Point 6: The "TESSA" acronyms refers exactly to "Situational Sea Awareness" while, broadly speaking, the actual research domain defines "SSA" as "Sea Situational Awareness".

projection of their status in the near future". Originally, Situation Awareness was an aviation term, but it has also been applied in many other safety critical domains (Stanton at al., 2001), like the maritime sector is.

Within TESSA, we meant\_SituationalSea\_SituationalAwareness as the capability to provide information about present and future sea conditions to support people during their activities at sea. This issue\_topic is strategically important for safer or

5

- optimal navigation, search and rescue, the assessment of the good environmental status of the marine ecosystem and the management of the sea territory. The lack of knowledge and awareness about sea conditions reduces readiness for reacting to emergencies and protecting the marine environment, with large socio-economic impacts and damages. With the "Marine Knowledge 2020" communication (Green Paper, 2012), the European Commission outlined the importance of marine knowledge in helping EU Member States meet targets about employment, innovation, education, social inclusion and combat
- 10 climate change as outlined in the "Europe 2020" strategy. Marine knowledge would provide the fundamental knowledge to facilitate the growth of a sustainable, job-creating "blue economy" in marine and maritime sectors by improving the competitiveness and efficiency of industry, public authorities and researchers.

It has been estimated that marine data fragmentation and inaccessibility causes a loss of 300 million Euros per year, and that estimation does not take into account the future growth in marine economy and the subsequent increased demand for data.

15 Opening up marine data allows new operators to enter the market while cross-systems interoperability allows businesses and academics to develop new products and services based on data from different sources and of different types. The impact assessment could be of the order of 200 million Euros per year. In addition to this, uncertainty is one of the main problem of those responsible for designing offshore structures, for managing fish stocks or for designing protected marine areas. In example, it has been estimated that a 25% reduction in uncertainty in future sea level rise would allow\_public authorities responsible for coastal management to save approximately 100 million Euros per year.

The TESSA project was born from the collaboration between advanced oceanographic research, scientific computing and information technology groups. The project aimed at developing advanced technological platforms (web-based, multi-target and multi-channel) for the production and provisioning of detailed information about sea conditions, at different complexity

- 25 levels, and creating decision Decision support Support systems (DSS) for ship routing, extreme conditions early warning, environmental quality, oil spill movement forecasts and search and rescue operations. Therefore, such a wide and complex objective from the scenario point of view has been faced by designing and developing a unique technological platform able to provide information services about sea conditions to multiple, domain-specific applications.
- 30 This paper describes the main driving factors of the system and the most innovative solutions that have been implemented in order to satisfy the outlined requirements. Section 2 describes the motivations and objectives behind this system; section 3 introduces related work in similar fields; section 3-4 presents the general architecture, including frontend and map-related infrastructure, while sections 54 describes the data analysis sub-system. Section 5-6 provides insights about the issues faced

2

**Commento [MS4]:** Review #1 – Point 6: follow up.

**Commento [MS5]:** All reviews: better context for motivations and related work sections. in implementing a scalable platform, with particular emphasis on the map rendering issues. Finally, section 67 outlines conclusions and future works.

**2 Motivations and objectives**

[revised manuscript text omitted]

**3 Related work**

Search for related or similar works about operational oceanography and sea situational awareness hasidentified two broad research and operational fields. The first deals with the development and operation of services for intermediate users,

- 20 individuals or organizations that need raw data in order to perform value-added analysis. In order to do that, their activities are centered on designing and developing platforms able to produce raw data (observational or output of forecasting models) and make them available to users, like research organizations, so that they can perform further processing anddeliver new services for other end-users. Studies in this context include (Bahurel et al., 2009) and (Moussat et al., 2016), describing development of hardware infrastructure and software appliances for collecting data and making them available for discovery
- 25 (e.g., a metadata service catalogue) and download. The second kind of research field is focused on the design and the development of services directly oriented to create and deliver value for end users. Projects in this field generally support well-defined scenarios, like oil spill (Zodiatis et al, 2016) or ship routing (Mannarini et al., 2013). In those cases, standalone clients (like mobile Apps) or dynamic web sites are created and maintained in order to allow the stakeholders to use the service.
- 30 From a technological point of view, one of the core activities within the TESSA Project, has been the development of a common platform for runningservices and decision support systems. This base infrastructure has been designed to be

**Commento [MS7]:** Review #1 – Point 7: Follow up (see other comment for more details)

extended and to support a variety of complex and different scenarios, like weather and marine forecast, ship routing, extreme conditions early warning, environmental quality, oil spill movement forecasts and search and rescue operations. Therefore, the novelty of the TESSA Project (and, in turn, the SSA Platform) resides in the creation of a shared infrastructure designed to be extended by additional applications, either produced in the context of the project itself or even

by other third parties. More specifically, the DSS that have been developed have taken full advantage of the existing ecosystem (which provides support for common concerns like security and specific map services), speeding up the development of each single service while increasing the total value of the SSA platform itself. This enables the platform to act as asingle entrypoint that professional and non-professional users can use in order to benefit of a collection of highly focused services, each dealing with a different aspect of the Sea Situational Awareness.

**10 3-4General Architecture**

20

25

Within the TESSA architecture, three main tiers\_can be identified (Figure 1): the client tier (e.g., mobile devicesnative Apps downloaded from the respective mobile platform's store or web pages running within browsers' windows) representing client applications (like SeaConditions), the Complex Data Analysis Module (CDAM) tier, hosting forecast and DSSspecific models and the SSA Platform, processing the forecast data from the CDAM tier and provisioning map data to the

15 client tier. The SSA platform communicates with the CDAM by fetching weather and marine forecast data as needed and by managing job submissions on behalf of the DSS applications.

In a glance, the SSA Platform provides the building blocks for creating SSA applications, like standard services (e.g., the static and dynamic map services), hosting infrastructure (e.g., web server and messaging support), multi-channel availability (based on standard Internet protocols) and personalized user experience (e.g., sharing user preferences across several applications and channels).

In order to make the data flow serviceable, there are several specialized software components involved, each one performing a well-defined task within the processing chain. These components\_can be grouped in two macro-areas: (1) front-end components, hosting applications and user-centered data, and (2) map and data analysis components, dealing with serving maps and querying data.

Frontend components include a web portal, which hosts Decision Support Systems (DSS)the server and web client UIs of DSS applications-like SeaConditions and VISIR; a message broker which enables communication between the DSS and their CDAM counterparts; and a user database hosting user preferences (like application settings and favourite places).

30 Map components include a download daemon which fetches data from the CDAM; a batch rendering system which performs initial ingestion within the system; a map service featuring tiled maps for forecast data; and a map server serving static maps like bathymetry. Data analysis is provided by software modules that allow for on-the-fly queries of forecast data, according **Commento [MS8]:** All reviews: better context for motivations and related work sections.

Commento [MS9]: Review #2 – Figure 1 is now explicitly mentioned in the text. Commento [MS10]: Review #1 – Point

7: clarified that "mobile devices" means native apps, which are different class of applications from rich web pages. Of course, this does not mean that web applications cannot be used from mobile browsers: native apps provide just more advanced user experiences and better integration with specific platform's features not available to conventional web applications (e.g., animations).

to a variety of patterns. The greater part of these components has been developed ad hoc for TESSA, leveraging Open Source software stacks and open Internet-based protocols wherever possible in order to maximize reuse of well-known and proven base packages.

---

## Author Comment (AC2) · 30 Sep 2016

[revised manuscript text omitted]

**3 Related work**

Search for related or similar works about operational oceanography and sea situational awareness hasidentified two broad research and operational fields. The first deals with the development and operation of services for intermediate users, individuals or organizations that need raw data in order to perform value-added analysis. In order to do that, their activities are centered on designing and developing platforms able to produce raw data (observational or output of forecasting models) and make them available to users, like research organizations, so that they can perform further processing anddeliver new services for other end-users. Studies in this context include (Bahurel et al., 2009) and (Moussat et al., 2016), describing development of hardware infrastructure and software appliances for collecting data and making them available for discovery (e.g., a metadata service catalogue) and download.

The second kind of research fieldis focused on the design and the development of services directly oriented to create and deliver value for end users. Projects in this field generally support well-defined scenarios, like oil spill (Zodiatis et al, 2016) or ship routing (Mannarini et al., 2013). In those cases, standalone clients (like mobile Apps) or dynamic web sites are created and maintained in order to allow the stakeholders to use the service.

From a technological point of view, one of the core activities within the TESSA Project,has been the development of a common platform for runningservices and decision support systems. This base infrastructure has been designed to be

extended and to support a variety of complex and different scenarios, like weather and marine forecast, ship routing, extreme conditions early warning, environmental quality, oil spill movement forecasts and search and rescue operations.

Therefore, the novelty of the TESSA Project (and, in turn, the SSA Platform) resides in the creation of a shared infrastructure designed to be extended by additional applications, either produced in the context of the project itself or even by other third parties. More specifically, the DSS that have been developed have taken full advantage of the existing ecosystem (which provides support for common concerns like security and specific map services), speeding up the development of each single service while increasing the total value of the SSA platform itself. This enables the platform to act as asingle entrypoint that professional and non-professional users can use in order to benefit of a collection of highly focused services, each dealing with a different aspect of the Sea Situational Awareness.

**3 4 General Architecture**

Within the TESSA architecture, three main tiers can be identified (Figure 1): the client tier (e.g., native Apps downloaded from the respective mobile platform's store or web pages running within browsers' windows) representing client applications (like SeaConditions), the Complex Data Analysis Module (CDAM) tier, hosting forecast and DSS-specific models and the SSA Platform, processing the forecast data from the CDAM tier and provisioning map data to the client tier. The SSA platform communicates with the CDAM by fetching weather and marine forecast data as needed and by managing job submissions on behalf of the DSS applications.

In a glance, the SSA Platform provides the building blocks for creating SSA applications, like standard services (e.g., the static and dynamic map services), hosting infrastructure (e.g., web server and messaging support), multi-channel availability (based on standard Internet protocols) and personalized user experience (e.g., sharing user preferences across several applications and channels).

In order to make the data flow serviceable, there are several specialized software components involved, each one performing a well-defined task within the processing chain. These components can be grouped in two macro-areas: (1) front-end components, hosting applications and user-centered data, and (2) map and data analysis components, dealing with serving maps and querying data.

Frontend components include a web portal, which hosts the server and web client UIs of DSS applications ; a message broker which enables communication between the DSS and their CDAM counterparts; and a user database hosting user preferences (like application settings and favourite places).

Map components include a download daemon which fetches data from the CDAM; a batch rendering system which performs initial ingestion within the system; a map service featuring tiled maps for forecast data; and a map server serving static maps like bathymetry. Data analysis is provided by software modules that allow for on-the-fly queries of forecast data, according

**Commento [MS8]:** All reviews: better context for motivations and related work sections.

**Commento [MS9]:** Review #2 – Figure 1 is now explicitly mentioned in the text.

**Commento [MS10]:** Review #1 – Point 7: clarified that "mobile devices" means native apps, which are different class of applications from rich web pages. Of course, this does not mean that web applications cannot be used from mobile browsers: native apps provide just more advanced user experiences and better integration with specific platform's features not available to conventional web applications (e.g., animations).

to a variety of patterns. The greater part of these components has been developed ad hoc for TESSA, leveraging Open Source software stacks and open Internet-based protocols wherever possible in order to maximize reuse of well-known and proven base packages.

[Figure]

[Figure]

**Commento [MS11]:** Review #1 – Point 2 (1) – Redesigned the figure to better show the interactions within the SSA Platform and how DSS architecture fits in.

**Figure 1 SSA Platform architecture (color key: blue for frontend components, green for map components, red for external modules or externally produced data, brown / orange for the DSS-specific components and interactions)**

5    The Web Portal is also tasked with ensuring the right authorizations, so that users can access applications according to their given permissions. Frontend components are made available through an HTTP reverse proxy[3] which manages URL mapping from a single host name (e.g., www.sea-conditions.com) to the right service, like the forecast webservice, the GeoServer[4], the message broker and the web portal itself.

The Map service provides the HTTP endpoints for serving forecast and observational maps and related metadata (though a

10   custom RESTful (Fielding, 2000) API[5], using standard protocols like WMTS and TMS. Each day, data within the SSA Data Storage are processed by a Map Pipeline which partially processes the most time-consuming map tiles: at runtime, the
* * *
[3]A reverse proxy is a web server fetching resources from other servers on behalf of some client: this is often used in IT departments in order to provide a unique facade (e.g., host name) which hides several other servers (as it happens within the TESSA architecture), thus improving systems changeability.
[4]http://geoserver.org/
[5]Application Program Interface, a collection of programmatic interfaces used for developing software applications.

system will autonomously render tiles that have not been built earlier while still serving already available ones. In addition to serving tiled maps, the Map service also contains logic for querying raw data (like the series of values of the sea temperature across a 4 days' timespan); an additional module, called ANSWER (detailed in a specific section), provides more complex data query features. An instance of GeoServer[6], a well-known software package for managing static maps in open formats, is also present in order to serve static map layers like bathymetry.

It must be underlined that, while the web user interface of each DSS application is being hosted within the portal, the computation-intensive part is not. In fact, the DSS are, by design, split in two parts between: the user interface and the module running the algorithm, the latter being hosted within the CDAM tier. This deployment scheme is functional for reusing the computational resources of the CDAM cluster for running model software while freeing the SSA Platform for performing its own data analysis and map-related tasks. Splitting each DSS is dictated by the different requirements of each application, where actions like submitting a request and getting a response may involve several seconds or minutes. The Message Broker makes easier to build application with such detached "submit-and-wait" logic: applications only have to assemble a message with their custom payload and queue them; the Message Broker will then track the request and ensure that the response from CDAM is then delivered to the client.

**34.1 Frontend components**

Frontend components deal with providing infrastructure services that allow hosted services to provide data to clients according custom web APIs. Decision Support Systems (DSS) are applications hosted within the portal while an HTTP Reverse-proxy hides the complexities of the underlying subsystems behind URLs based on a single hostname (like it happens for www.sea-conditions.com). Within TESSA, DSS applications share a common client-server architectural pattern by defining a client-hosted UI (native App or web page) interacting with a server counterpart (e.g., RESTful webservice). Thus, the SSA Platform provides an infrastructure that supports this architecture (like reusable authentication and messaging components that can be re-used) and makes their implementation easier.

The Web portal is a web container, based on Liferay[7] technology and supporting the latest JSR Portlet JSR 286[8] standard, customized for the requirements of the TESSA project and hosting the applications, their web APIs and providing fundamental shared services like authentication, authorization and user profile and preferences management.

Frontend applications, like all DSS, are hosted within the portal as standard portlets, min-web applications that can benefit of the infrastructure provided by portal containers like security, data sources, and so on. Within TESSA, DSS applications, like

**Commento [MS12]:** Review #1 – Point 3 – A better definition of DSS application is provided in order to reduce confusion when speaking about DSS, DSS application and clients. Within this paper, the same name may be used (e.g., SeaConditions) for referring to the UI or the whole stack: when not clear from the context, a more explicit wording is used.
* * *
[6]http://geoserver.org/

[revised manuscript text omitted]

* * *
[9] Central Processing Unit, the execution unit for software instructions.

**Commento [MS13]:** Review #1 – Point 8 (1) – fixed typo

**Commento [MS14]:** Review #1 – Point 8 (4) – fixed wrong section reference

The solution consisted in implementing a distributed rendering model: a master node navigates the tile pyramid and generating all rendering tasks. The latter are then queued to a rendering grid composed of worker nodes that only have one single job: to render as many tiles as they can according to their CPU cores. Scalability is then achieved by adding as many worker nodes as required and then automatically made available to the system, without a restart being required; similarly, a node may be turned off without the rendering cluster falling apart, besides having less computational power available. Each worker node accesses data from SSA Platform Data Storage, executes the rendering (employing different techniques as required) and returns the result (consisting of image bytes plus metadata). Tasks are defined by descriptors assembled by clients (e.g., the dynamic maps service or the batch rendering system) and queued to the cluster. A built-in load balancing feature ensures that tasks are dispatched in a roughly fair manner across different nodes, so that no single node is particularly overloaded. We integrated the Hazelcast[10] grid computing middleware, which provides infrastructure for coordinating tasks among multiple processes and machines and built on top of it our rendering pipeline.

Operational running of the system is allowed by the means of system logs and mailing: when issues happen (e.g., unavailability of data, invalid or corrupted files, internal errors, and others), personnel can investigate the problem and operate in order to fix it (like, re-running rendering jobs for variables that were not available at an expected time). Supporting operational teams has been crucial in order to improve the final availability of the platform services to the final users and we further think of improving it by collecting statistics in order to formulate realistic  Service  Level  Agreements (SLA) with external users.

**Commento [MS15]:** Review #1 – Point 8 (3) – Capitalized letters for SLA

**4.3 Physical deployment**

The aforementioned components are distributed across several nodes (virtual machines), with different hardware configurations (CPU, memory and disk capabilities)and the same Operating System (Ubuntu Linux[11] 14.04). In the current iteration, there are four types of nodes, each hosting one or more components of the SSA Platform: Portal Node, Database Node, File Store Node and Worker Node.
The Portal Node hosts Liferay, the Message Broker and the DSS (web APIs and UIs); the Database Node hosts the MySQL instance with portal's and users' data; the File Store Node hosts the download daemon and provides storage for the NetCDF files in addition to running the components belonging to the Map Service (including the GeoPortal) while the Worker Node implements the rendering logic.Both the Portal, Database and FileStore nodes have 4 CPU cores and 16 Gigabytes (the File Store has 200 Gigabytes of additional disk space for storing the most recent downloaded data).
* * *
[10]http://hazelcast.org/
[11]http://www.ubuntu.com/

Each Worker Node (with 8 Gigabytes of RAM, 4 CPU core and about 30 Gigabytes of disk space) hosts a single rendering instance of the Map Service, each of them able to plot a number of tiles equal to the number of CPU cores (that is, four concurrent rendering operations for each Worker Node). New Worker Nodes can be added easily in order to increase computational capacity and, collectively, all Worker Nodes cooperate to share the workload generated by the Batch Map Pipeline and the on-the-fly rendering requests coming from the Map Service. The bigger the number of map tiles to plot, the larger the required worker resources need to be (see section 6 for more information about the rendering process): the current rendering grid infrastructure scales linearly with the number of Worker Nodes by processing map tiles in parallel. Thread-level parallelism requires CPUs with multiple cores while process-level parallelism (e.g., multiple nodes) can combine multiple instances hosted on different machines. While thread-level parallelism is somewhat more efficient than process level (e.g., sharing data within a single process has less overhead that sharing across different processes and/or machines),machines with many cores can become exceedingly expensive while combining simpler and cheaper machines becomes paramount. Therefore, the architecture is potentially cheaper to operated and extend than single-process solutions: Google uses the same basic concept for increasing the throughput of their search engine in an affordable way (Dean et Al, 2004).

**Commento [MS16]:** Review #2 – Point 4 (2) – Clarified how software components are distributed to physical nodes.

**45Data analysis**

In addition to map rendering, the SSA platform also provides services for querying the data, currently providing a sequence of environmental values for a given variable. As improvement to data analysis, a sub-system (called ANSWER) for searching time slices matching specific conditions has been implemented (e.g., winds and sea currents matching some externally provided thresholds) and made available to external systems (like MySeaConditions (Coppini, 2016)).

**45.1 Data queries and statistics**

The simplest data query involves fetching values for a given variable from each NetCDF file, a process that may require a few seconds before completing, depending on the current system load and complexity of the involved computations, which depend on the particular variable. In order to speed the entire computation up, a data files indexing is performed and consulted for checking each file: read is performed using Unidata's NetCDF-Java library[12] once the correct file has been identified.

The concept of a tile as a pre-defined geo-referenced bounding box has been used for data analysis too: these "data tiles" can be used for relatively fast execution of statistics computations (like average, standard deviation, minimum and maximum values). Web APIs, either custom TESSA-defined or OGC's standards, make these features available to external clientsapplications, like SeaConditions. A significant limitation of the current implementation is that data queries are performed sequentially, taking no advantage of the rendering grid: this is something that we plan to improve in the future

**Commento [MS17]:** Review #1 – Point 3 – Used the term "Application" instead. SeaConditions is both the whole application stack (client and server) and the name of the native app available on the store. Of course, users interacts with the client part but the whole stack is used in providing answers to their requests.

[revised manuscript text omitted]

P. Bahurel, F. Adragna, M. J. Bell, F. Jacq, J. A. Johannessen, P. Le Traon, N. Pinardi, J. She: Ocean Monitoring and forecasting core services, the European MyOcean example, Proceedings of OceanObs'09, 2009

E. Moussat, N. Pinardi, G. Manzella, F. Blanc: EMODnet MedSea Checkpoint for sustainable Blue Growth, EGU General Assembly, 2016

5   G. Zodiatis, M. De Dominicis, L. Perivoliotis, H. Radhakrishnan, E. Georgoudis, M. Sotillo, R.W. Lardner, G. Krokos, D. Bruciaferri, E. Clementi, A. Guarnieri, A. Ribotti, A. Drago, E. Bourma, E. Padorno, P. Daniel, G. Gonzalez, C. Chazoti, V. Gouriou, X. Kremer, S. Sofianos, J. Tintore, P. Garreau, N. Pinardi, G. Coppini, R. Lecci, A. Pisano, R. Sorgente, L. Fazioli, D. Soloviev, S. Stylianou, A. Nikolaidis, X. Panayidou, A. Karaolia, A. Gauci, A. Marcati, L. Caiazzo, M. Mancini: The Mediterranean Decision Support System for Marine Safety dedicated to oil slicks predictions, Deep-Sea Research Part II:

10  Topical Studies in Oceanography, 2016

G. Mannarini, G. Coppini, P. Oddo, N. Pinardi: A Prototype of Ship Routing Decision Support System for an Operational Oceanographic Service, TransNav, 2013

R. Fielding, Architectural Styles and the Design of Network-based Software Architectures, Ph.D. Dissertation, University of Irvine, 2000

15  G. Coppini, P. Marra, R. Lecci, N. Pinardi, S. Cretì, M. Scalas, L. Tedescoet Al., SeaConditions: present and future sea conditions for safer navigation, Natural Hazards and Earth System Science, in this issue, 2016

Tarquini, S. et Al., "Immagini di modelli digitali a medio/alta risoluzione navigabili via web: un esempio di condivisione di banche dati geografiche di grandi dimensioni tramite Google Earth", Istituto Nazionale di Geofisica e Vulcanologia, Rapporti Tecnici INGV n. 73, 2008

**Commento [MS27]:** All reviews: added references for "Related work" section.